# How to Account for Changes in Carbon Storage from Coal Mining and Reclamation in Eastern China? Taking Yanzhou Coalfield as an Example to Simulate and Estimate



Jiazheng Han [1], Zhenqi Hu [1,2,3,*], Zhen Mao [1], Gensheng Li [4], Shuguang Liu [1], Dongzhu Yuan [2] and Jiaxin Guo [2]

1   School of Environment Science & Spatial Informatics, China University of Mining & Technology, Xuzhou 221116, China; cumthjz@cumt.edu.cn (J.H.); maozhen@cumt.edu.cn (Z.M.); liusg0925@cumt.edu.cn (S.L.)
2   School of Geosciences & Surveying Engineering, China University of Mining & Technology (Beijing), Beijing 100083, China; tbp150202004z@student.cumtb.edu.cn (D.Y.); bqt1900204046@student.cumtb.edu.cn (J.G.)
3   Jiangsu Key Laboratory of Coal-Based Greenhouse Gas Control and Utilization, China University of Mining and Technology, Xuzhou 221008, China
4   School of Public Policy & Management, China University of Mining & Technology, Xuzhou 221116, China; tb19160023b0@cumt.edu.cn
*   Correspondence: huzq@cumtb.edu.cn

**Abstract:** Carbon sequestration in terrestrial ecosystems plays an essential role in coping with global climate change and achieving regional carbon neutrality. In mining areas with high groundwater levels in eastern China, underground coal mining has caused severe damage to surface ecology. It is of practical significance to evaluate and predict the positive and negative effects of coal mining and land reclamation on carbon pools. This study set up three scenarios for the development of the Yanzhou coalfield (YZC) in 2030, including: (1) no mining activities (NMA); (2) no reclamation after mining (NRM); (3) mining and reclamation (MR). The probability integral model (PIM) was used to predict the subsidence caused by mining in YZC in 2030, and land use and land cover (LULC) of 2010 and 2020 were interpreted by remote sensing images. Based on the classification of land damage, the LULC of different scenarios in the future was simulated by integrating various social and natural factors. Under different scenarios, the InVEST model evaluated carbon storage and its temporal and spatial distribution characteristics. The results indicated that: (1) By 2030, YZC would have 4341.13 ha of land disturbed by coal mining activities. (2) Carbon storage in the NRM scenario would be 37,647.11 Mg lower than that in the NMA scenario, while carbon storage in the MR scenario would be 18,151.03 Mg higher than that in the NRM scenario. Significantly, the Nantun mine would reduce carbon sequestration loss by 72.29% due to reclamation measures. (3) Carbon storage has a significant positive spatial correlation, and coal mining would lead to the fragmentation of the carbon sink. The method of accounting for and predicting carbon storage proposed in this study can provide data support for mining and reclamation planning of coal mine enterprises and carbon-neutral planning of government departments.

**Keywords:** carbon storage; high groundwater level mining area; land reclamation; mining subsidence prediction; land use simulation

## 1. Introduction

The climate problem characterized by global warming has attracted wide attention from the international community, which has brought enormous challenges to the development of human society, the economy, and the world energy structure. Global warming of the past 150 years has exceeded total global warming of the past 6500 years [1]. Increased greenhouse gas concentrations due to human disturbance, such as rapid industrialization,

overexploitation and utilization of fossil fuels, and land use change, are considered the leading causes of significant warming, with carbon dioxide accounting for 70% of total atmospheric greenhouse gas emissions [2]. As a critical carbon pool in the carbon cycle, terrestrial ecosystems play an important role in maintaining the global carbon cycle. Increasing terrestrial ecosystem carbon storage can effectively reduce atmospheric carbon dioxide content [3]. Timely and practical assessment of terrestrial ecosystem carbon storage is significant to global/regional carbon cycle research, mitigation of climate change, and sustainable development.

Carbon storage has been estimated at different scales for forests [4,5], cities [6–8], cultivated land [9,10], wetlands [11,12], and other ecosystems. At present, the leading carbon storage estimation methods include measurement methods [13,14], inventory-based methods [15,16], estimation methods based on remote sensing data [17–19], etc. The measurement method needs to collect samples on the spot and take them back to the lab to measure carbon density. Estimating carbon storage in a small area is more accurate but time-consuming and laborious. Inventory-based methods are often used to estimate forest carbon stocks. Data on tree species, area, age, and height are needed for calculation. Remote sensing technology can play a key role in estimating carbon storage's temporal and spatial changes in the study area with a large area and complex surface ecology. Net primary productivity (NPP) [20,21], land use and land cover (LULC) [22,23], and other data or related forms are usually used in this context. The combination of the InVEST model and land use change simulation model can predict the future carbon storage pattern in the study area to carry out ecological conservation measures in advance and improve the carbon sequestration capacity of the regional ecosystem [24,25]. In recent years, various land use simulation models have been proposed [26–28]. To improve the simulation accuracy, it is necessary to comprehensively consider climate, transportation, population, economy, and other factors based on the law of changing land use. Liang [29] proposed a patch-generating land Use Simulation (PLUS) model, which can better explore the incentives of various land use changes and simulate the patch-level changes of various types of land use.

In recent decades, China's rapid economic development has been accompanied by increasing energy demand, with coal accounting for 56.8% of the country's total energy consumption in 2020. Coal will continue to be China's primary energy source for some time to come. The large-scale exploitation of coal resources has caused large-scale damage to the surface and has affected the original ecosystem function, including the carbon storage function [30,31]. Some scholars have studied carbon storage in the mining area [32–35]. However, coal mining is a dynamic process, and the existing research does not estimate the future changes in carbon storage based on the analysis of the impact of mining on the surface. And these studies also do not explain the contribution of land reclamation to increasing carbon storage.

Especially in the mining area with a high groundwater level in east China, coal is mainly mined underground, which results in severe subsidence and damage to the surface and to the formation of a subsidence basin. The high groundwater table can easily rise above the surface elevation after the coal mining subsidence, forming perennial or seasonal water [36,37]. As a result, the original land production and living facilities suffers severe damage, soil salinization, and vegetation degradation, which seriously reduces the carbon sequestration function of the terrestrial ecosystem [38,39]. It is urgent to conduct ecological restoration work to improve carbon sequestration levels in eastern China mining areas.

China has set a target to achieve carbon neutrality by 2060. It is of great significance for China to increase carbon sinks by rapidly and reasonably estimating and predicting the impact of coal mining on the spatio-temporal change of carbon storage. Based on the prediction of coal mining subsidence, this study analyzes the changes and spatial layout of carbon reserves in mining areas under different development scenarios, so as to provide a reference for local governments and coal mining enterprises to carry out environmental governance. In this paper, using the Yanzhou coalfield (YZC) as an example, the main research objectives are:

(1) Propose a method to estimate the impact of mining and reclamation measures on carbon storage;
(2) Prediction of land change in mining areas affected by mining by coupling the PLUS model and probability integration method;
(3) Spatial–temporal relationship of carbon storage using the spatial analysis method.

## 2. Materials and Methods

### 2.1. Study Area

YZC (Figure 1) is in Jining, southwest of Shandong Province, China. It belongs to the Huai River basin and is relatively rich in water resources. Its inland rivers are all seasonal rivers, mainly including the Si River, the White Horse River, the Shahe River, the muddy river, etc. YZC is a transitional climate between temperate monsoon ocean and continental, with four distinct seasons and abundant rainfall. YZC is about 32.2 km in length from north to south, 23.4 km in width from east to west, with a total area of 403.7 km² and 14 coal mines, including Datong, Baodian, Yangcun, Tianzhuang, Taiping, Henghe, Liyan, Beiju, Nantun, Dongtan, Xinglong, Xingcun, Gucheng, and Shanjiacun. Coal seam thickness is generally 1–10 m, buried depth is below 300 m, mostly gently inclined coal seam, and the coal mines have adopted shaft development for underground mining.

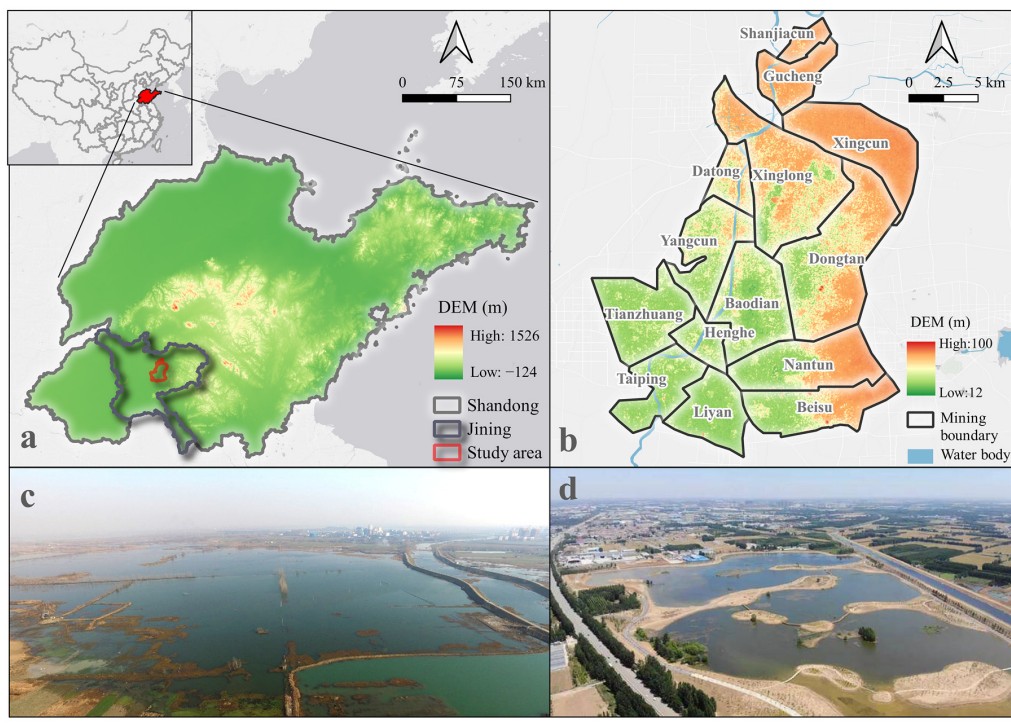

**Figure 1.** Schematic diagram of the study area: (**a**) location of YZC; (**b**) distribution of coal mines in YZC; (**c**) coal mining has led to water accumulation in large areas of cropland; (**d**) wetland landscape was formed after reclamation.

YZC is an important part of the Luxi coal base (one of China's fourteen large coal bases). In the meantime, YZC is a typical coal-grain complex area with more than 70% cropland. The coal–grain complex area is where the coal resources and the grain production space overlap. It bears the critical responsibility of national food security and mineral resources supply [40]. YZC entered a period of high-intensity mining from the late 1980s to the 1990s and is still operating normally today. Large cropland areas collapsed after decades of coal mining and then turned into water areas. Native terrestrial ecosystems have been severely disturbed and are difficult to repair, as shown in Figure 1c, which directly threaten regional and even national food security, energy security, and ecological security.

More and more attention has been paid to the ecological rehabilitation of mining areas in recent years. In 2016, the State Council General Office issued Opinions on accelerating the comprehensive control of coal mining subsidence areas, which laid out plans for controlling coal mining subsidence areas throughout the country. In 2020, Shandong province issued the Special plan for comprehensive treatment of coal mining subsidence, of which YZC is the key area. Under the background of carbon neutrality, the ecological restoration of the mining area is endowed with a new connotation and target. Therefore, this study takes YZC as an example to discuss the carbon sequestration effect of coal mining and restoration.

### 2.2. Data Source

The study used Landsat TM/OLI data from April to May in 2010 and 2020, downloaded from the USGS (https://earthexplorer.usgs.gov/) (accessed on 1 December 2021), for LULC classification. ENVI 5.3 was used for image preprocessing, including image cropping, radiation correction, and atmospheric correction. In addition, the coal mining data used in this study were provided by local enterprises, and the spatial data sources and descriptions for carbon storage assessment and land use simulation are shown in Table 1 and Figure 2.

**Table 1.** Description of spatial data.

| Data Name | Spatial Resolution | Sources |
|---|---|---|
| Soil type | 1 km | HWSD v1.2 |
| Precipitation Temperature | 1 km | WorldClimv2.1 |
| DEM | 30 m | SRTM1 v3.0 |
| GDP density Population density | 1 km | http://www.geodoi.ac.cn/ (accessed on 6 December 2021) |
| Road network information | - | https://www.openstreetmap.org/ (accessed on 6 December 2021) |
| Distribution of railway stations Government location | - | http://lbsyun.baidu.com/ (accessed on 6 December 2021) |

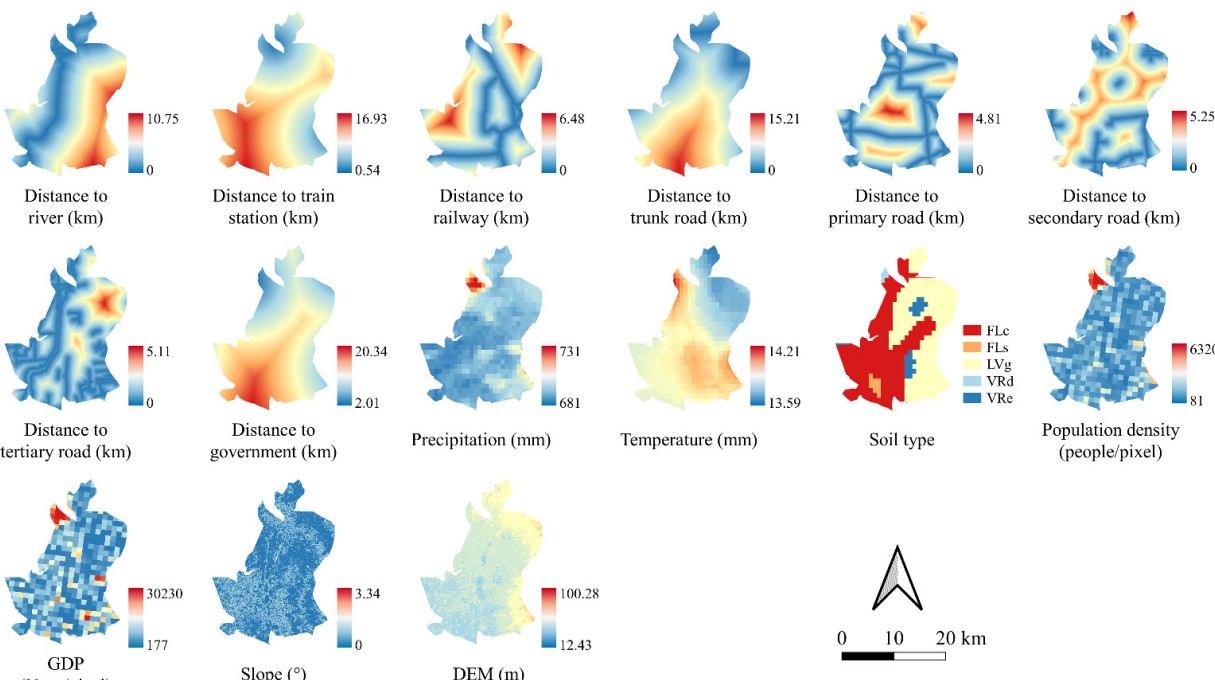

**Figure 2.** Spatial data for land use modeling.

### 2.3. Method

The analysis process of this study includes four key steps. First, according to the mining parameters of each mining area in YZC, the probability integral method (PIM) was used to predict the subsidence in 2030 and classify it. Second, combined with the subsidence situation and LULC in 2010 and 2020 and various influencing factors, the PLUS model was used to simulate land use in the three scenarios of 2030. Third, the carbon density was corrected, and the InVEST model was used to calculate the carbon reserves of each year and each mine to calculate the carbon effect of coal mining and land reclamation. Fourth, ESDA was used for spatial analysis of the gridded carbon density. The organizational process of this study is shown in Figure 3.

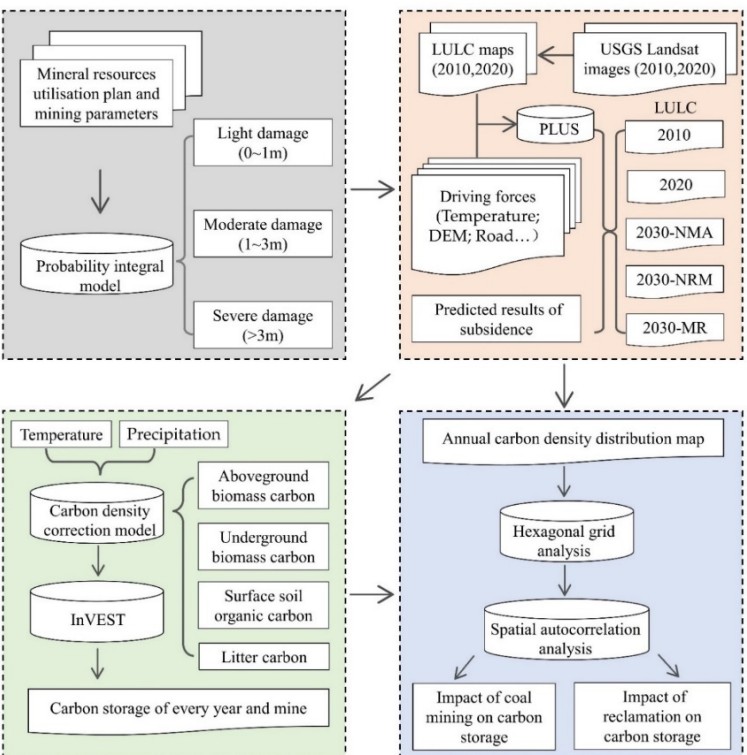

**Figure 3.** Research flowcharts.

### 2.3.1. Principle of Predicting Mining Subsidence

By predicting mining subsidence, we can study the time and space distribution law of strata and surface movement affected by the mining and master the situation of land subsidence in the mining area to take the corresponding disaster prevention and reclamation work.

PIM is the most widely used method for predicting mining subsidence in China, which is also one of the methods recommended by the National Energy Administration in its 2017 revision of Regulation of mining and pillar leaving under building, water-body, railway and main underground engineer [41]. PIM is based on stochastic medium theory, which decomposes the whole mining area into an infinite number of small units. The influence of the entire mining process on the strata and the surface is equal to the sum of the influence of each mining unit on the strata and the surface. The subsidence basins of surface units caused by the unit mining process show normal distribution, consistent with the distribution of probability density [42–44]. Thus, the equation for the subsidence profile generated by the entire mining process can be expressed as an integral formula for the probability density function:

$$W_e(x) = \frac{1}{r^2} \exp\left(-\pi \frac{x^2}{r^2}\right) \tag{1}$$

$W_e(x)$ is the unit point subsidence value, $r$ is the main influence radius, mainly related to the unit mining depth and the primary influence angle, and $x$ is the horizontal coordinate value of any point on the surface.

The movement of the overlying strata after coal mining does not reach the surface immediately, and the process is gradual and relatively slow. When the coal mining is completed for some time, the movement of the surface passes through the initial stage, the active stage, and the declining stage. Eventually, a steady state will emerge, as shown in Figure 4. All the subsidence areas obtained in this study are stable.

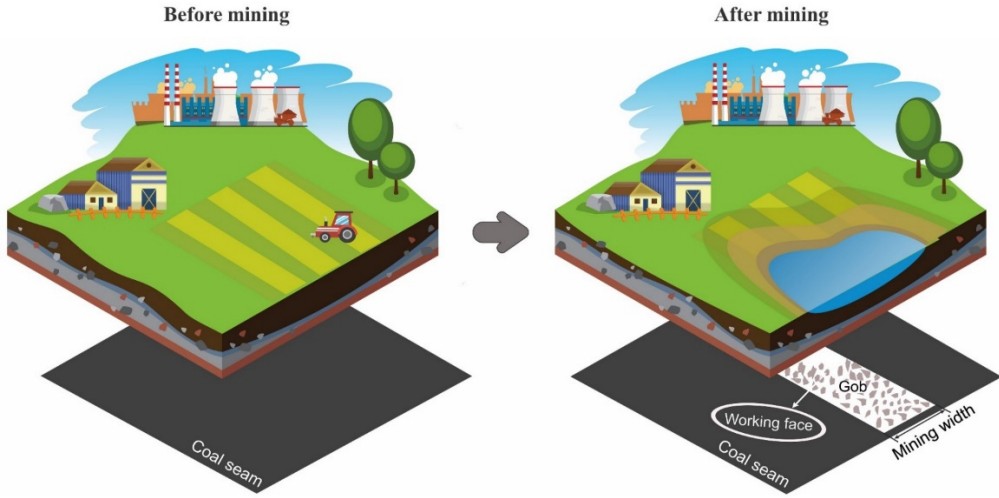

**Figure 4.** Schematic diagram of surface ecosystem disturbance caused by underground mining.

According to the relevant provisions of Comprehensive land remediation norms, combined with the actual mining area, the degree of land damage of coal mining subsidence is divided into three types. Surface vertical subsidence within 1 m (including 1 m) is light damage, 1–3 m (including 3 m) is moderate damage, and more than 3 m is severe damage [45]. According to the mining plan, mineral resources utilization plan, and mining parameters provided by each coal mine (corner coordinates, coal seam strike, coal seam mining thickness, mining depth, coal seam dip angle, subsidence coefficient, main influence angle tangent, horizontal movement coefficient, and influence propagation angle), we used MSPS software based on PIM for predicting coal mining subsidence in 2030.

### 2.3.2. Classification and Simulation of LULC

Random forest (RF) is an integrated classification learner that classifies samples based on multiple decision trees [46]. The basic composition of RF is a decision tree, trained and classified by one hundred/thousand decision trees. Finally, a decision is made by considering the voting results of multiple learners. Compared with the traditional decision tree method, RF integrates multiple classifiers, which makes up for the defect of a single classifier. The RF classifier has many advantages over other algorithms using remote sensing data to produce land cover change images [47]. Compared with most algorithms, including advanced algorithms such as support vector machine (SVM), RF algorithms are more efficient and require fewer parameters to input [48]. Therefore, this paper used the RF classification tool of ENVI5.3 to classify the LULC of YZC with the object-oriented method. According to the classification of LULC in the Chinese Academy of Sciences [49] and the requirements of China's Guidelines for the preparation of mine geological environment protection and land reclamation plans for the classification of mining areas, the land use types of YZC were classified into six types: cropland (paddy and dry farmland), woodland (deciduous forest land, evergreen forest land, mixed forest land), grassland (typical grassland and shrub forest land), built-up land (construction land and traffic land), water body (lake, river, reservoir, pond, marsh, and wetland) and unused land (exposed

land and unused land). The classification accuracy was evaluated by confusion matrix, producer accuracy (PA), user accuracy (UA), overall accuracy (OA), and Kappa coefficient.

The patch-generating land use simulation (PLUS) model, which integrates land expansion strategy analysis (LEAS) and a CA model based on multi-type random patch seeds (CARS), was proposed in 2021. Compared with other land use simulation models, the two sub-modules of PLUS have outstanding advantages. LEAS absorbs the advantages of the traditional transformation analysis strategy and can better explore the driving factors of land use change. Conversely, CARS combines random seed generation and the threshold decline mechanism, which can better simulate the evolution of land use patch level. At present, the PLUS model has been proven to be a more effective model, which provides more accurate simulation results [29,50]. Therefore, this study uses this model for land use simulation.

At present, the local mining arrangements of YZC are up to about 2030, and the ecological restoration planning period issued by the management department is also up to 2030. Therefore, 2030 was taken as the year of land use simulation in this paper. The study used the YZC LULC data for 2010 and 2020; the LEAS model to analyze the impacts of climate, population, GDP, and transport on land expansion LULC in 2030 was simulated by the CARS module.

To evaluate the different impacts of future mining disturbance and land reclamation on YZC land use and then calculate the carbon effect under different policy tendencies, there are three scenarios for the 2030 development of the study area:

(1) No mining activities (NMA) scenario: It is assumed that YZC will have no mining activity until the end of 2030. The law of land use conversion is determined by the LEAS module, but the water area is no longer increasing. This scenario serves as a benchmark land use scenario to measure the impact of mining and reclamation activities on the land.

(2) No reclamation after mining (NRM) scenario: Because of the small subsidence extent, the lightly damaged areas are easy to restore to the original type, while the moderately damaged areas are relatively deep in subsidence and are easy to transform from the original type into the water area.

(3) Mining and reclamation (MR) scenario: The lightly damaged area is reclaimed to the original type of land. As the surface of the mining area is mainly cropland, to ensure food security, according to the principle of priority cropland, the scene of land transfer rules is set as follows: the built-up land and water area are restored to the original type of land in the moderately damaged area, other areas are restored to the cropland, the heavily damaged area is not easy to be reclaimed, and the land type is transformed into the water area.

NRM scenario and MR scenario were simulated based on the NMA scenario, as shown in Table 2.

**Table 2.** Land conversion rules in NRM and MR scenarios.

| | NRM | MR |
|---|---|---|
| Light damage | Natural restoration to the original LULC | Natural restoration to the original LULC |
| Moderate damage | All LULC will be converted to the water area | Built-up land and water will be restored to the original land. Other LULC will be reclaimed for cropland |
| Severe damage | All LULC will be converted to the water area | All LULC will be converted to the water area |

### 2.3.3. Calculation of Carbon Storage

The InVEST model, developed jointly by Stanford University, The Nature Conservancy, and the World Wide Fund for Nature in the United States, aims to simulate changes in the quantity and quality of ecosystem values under different land use scenarios, achieving a quantitative spatial assessment of ecosystem service values [51]. The carbon module in the invest model is a direct and effective model to evaluate the impact of land use change on

carbon storage. It has the advantages of convenient use, flexible parameters, and relatively accurate results and has been widely used [52,53]. Carbon storage (C) is determined by aboveground biomass carbon ($C_{above}$), underground biomass carbon ($C_{below}$), surface soil organic carbon ($C_{soil}$) and litter carbon ($C_{dead}$) in terrestrial ecosystems.

$$C = C_{above} + C_{below} + C_{soil} + C_{dead} \tag{2}$$

Many studies showed a significant linear relationship between carbon density and climate factors [54,55]. In 2018, based on 14,371 measured data, Tang [56] developed models of soil organic carbon, bioorganic carbon and litter carbon under different climatic conditions. Combined with the research of Alam [57] and Zhou [58], we used the relationship model (Formulas (3)–(11)) to modify the carbon density data of Ma [59] in the Yellow River Delta of Shandong province. By referring to the local annual statistical yearbooks, the average temperatures of Jining City, where YZC is located, and Dongying City, the central city of the Yellow River Delta, in the past decade are 14.9 and 14.0 °C, respectively, and the average precipitation is 715.6 and 593.4 mm, respectively.

Regression model considering temperature (MAP > 400 mm):

$$C_{ST} = -3.4 \times \text{MAT} + 157.7 \tag{3}$$

$$C_{BT} = -0.4 \times \text{MAT} + 43.0 \tag{4}$$

$$C_{DT} = -0.03 \times \text{MAT} + 2.03 \tag{5}$$

Regression model considering precipitation (MAT > 10 °C):

$$C_{SP} = 0.03 * \text{MAP} + 45.3 \tag{6}$$

$$C_{BP} = 0.02 * \text{MAP} + 5.87 \tag{7}$$

$$C_{DP} = 0.004 * \text{MAP} + 0.85 \tag{8}$$

where $C_{ST}$ and $C_{SP}$ are the fitted values of MAT and MAP for soil carbon density, $C_{BT}$ and $C_{BP}$ are the fitted values of MAT and MAP for biomass carbon density, respectively, $C_{DT}$ and $C_{DP}$ are the fitted values of MAT and MAP for carbon density of litter, respectively.

Correction model:

$$K_{SP} = \frac{C_{SP1}}{C_{SP2}} \; ; \; K_{ST} = \frac{C_{ST1}}{C_{ST2}} \; ; \; K_s = Average(K_{SP}, K_{ST}) \tag{9}$$

$$K_{BP} = \frac{C_{BP1}}{C_{BP2}} \; ; \; K_{BT} = \frac{C_{BT1}}{C_{BT2}} \; ; \; K_B = Average(K_{BP}, K_{BT}) \tag{10}$$

$$K_{DP} = \frac{C_{DP1}}{C_{DP2}} \; ; \; K_{DT} = \frac{C_{DT1}}{C_{DT2}} \; ; \; K_D = Average(K_{DP}, K_{DT}) \tag{11}$$

where $K_{SP}$ and $K_{ST}$ are the correction coefficients of soil carbon density considering precipitation and temperature factors, respectively, $K_{BP}$ and $K_{BT}$ are the correction coefficients of soil carbon density considering precipitation and temperature factors, respectively, $K_{DP}$ and $K_{DT}$ are the carbon density correction coefficients of the litter biomass considering the precipitation and temperature factors, respectively, 1 and 2 correspond to Dongying and YZC, respectively, and $K_S$, $K_B$ and $K_D$ were the revised coefficients of soil carbon density, biomass carbon density and litter biomass carbon density, respectively.

According to the modified model, the carbon density of different land use in YZC was determined, as shown in Table 3.

**Table 3.** Carbon density (Mg/ha) of different LULC in YZC.

|  | $C_{above}$ | $C_{below}$ | $C_{soil}$ | $C_{dead}$ |
|---|---|---|---|---|
| Cropland | 11.71 | 3.19 | 18.08 | 0.43 |
| Woodland | 36.40 | 7.88 | 19.40 | 2.99 |
| Grassland | 13.41 | 5.96 | 16.05 | 1.50 |
| Built-up land | 0.00 | 0.00 | 8.13 | 0.00 |
| Unused areas | 1.06 | 0.00 | 15.24 | 0.00 |
| Water area | 2.13 | 1.06 | 10.16 | 0.00 |

### 2.3.4. Spatial Analysis

The spatial and temporal characteristics of different areas of YZC carbon storage can be mined by grid analysis. The basic grid shapes commonly used in grid analysis include quadrilateral grid, hexagon grid, and triangle grid, etc. Hexagon mesh can reduce the sample deviation due to the boundary effect of the mesh shape [60]. Since hexagons have the same distance from the center of mass in all six directions, neighborhood analysis can yield more realistic results [61]. Therefore, according to YZC's actual situation, the hexagon grid with a height of 300 m was selected for the spatial analysis of carbon storage.

Exploratory spatial data analysis (ESDA) is an analytical method of data depth mining and spatial visualization. The core content is to measure the spatial correlation pattern through spatial autocorrelation analysis, and then measure the homogeneity or heterogeneity of spatial data. In this research, ESDA was implemented by QGIS 3.16 and GEODA1.6.7, and we chose the average carbon density of the cell grid as the ESDA variable. The spatial autocorrelation recognition is carried out using the Global Moran's I index and the Local Moran's I index. The global spatial autocorrelation is an index parameter to study the global spatial correlation and similarity of the attribute values of the adjacent grids in the studied region. The value range of Global Moran's I is [−1, 1]. When the value is positive, the spatial correlation is positive, and the larger the value is, the more pronounced the space correlation is. When the value is negative, the spatial correlation is negative. The smaller the value, the more pronounced the spatial correlation is. When the value is zero, the spatial distribution is random. Local Moran's I can measure the local spatial correlation and spatial differentiation between each grid and its surrounding grid.

$$I = \frac{\sum_{i=1}^{n} \sum_{j=1}^{n} w_{ij}(x_i - \overline{x})(x_j - \overline{x})}{S^2 \left( \sum_i \sum_j w_{ij} \right)} \tag{12}$$

$$I_i = \frac{(x_i - \overline{x}) \sum_{j=1}^{n} w_{ij}(x_i - \overline{x})}{S^2} \tag{13}$$

$$S^2 = \frac{1}{n} \sum_{i=1}^{n} (x_i - \overline{x})^2 \tag{14}$$

where $n$ is the total number of grid units; $x_i$ ($x_j$) is the measurement value of grid unit $i(j)$; $(x_i - \overline{x})$ is the deviation between the measured value and the average value on the $i$th grid element; $w_{ij}$ is the standardized spatial weight matrix; $S^2$ is variance.

## 3. Results

### 3.1. Area of Land Subsidence Expected

According to mining enterprises' mining plan and mineral resources planning, the MSPS software integrated with PIM was used to predict coal mining subsidence. The spatial distribution of subsidence is shown in Figure 5a. Because some mines are in the closed stage without mining damage, the subsidence location is mainly distributed in the middle of YZC. By 2030, YZC's subsidence area would reach 4341.13 ha, accounting for 10.74% of the total coalfield area. The total area of lightly land damage is 1484.66 ha, the

total area of moderate land damage would be 1275.08 ha, and the entire area of severe land damage would be 1581.38 ha. The Dongtan mine's most significant light and moderate land damage areas reached 637.47 ha (45.36%) and 532.28 ha (41.75%). In contrast, the mines with the most significant proportion of severe land damage are Baodian, reaching 491 ha (31%).

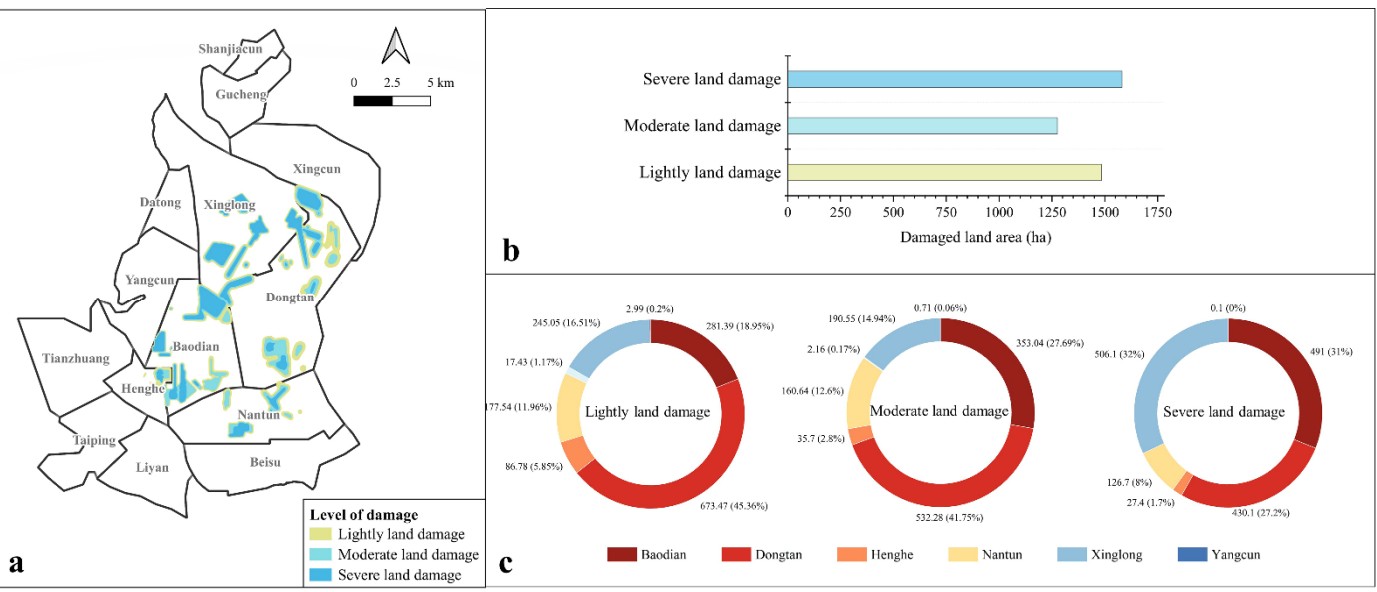

**Figure 5.** Predicting outcomes of land damage in 2030: (**a**) spatial distribution of damage; (**b**) area proportion of different coal mines with varying degrees of damage; (**c**) area statistics for various degrees of damage).

### 3.2. LULC Changes and Simulations

Using Google Earth Pro high-resolution images, we identified 1000 sample points as test samples and evaluated the accuracy of the classification results using the obfuscation matrix (shown in Table 4). In 2010 and 2020, the overall accuracies were 91.00% and 92.50%, respectively, and the kappa coefficients were 0.86 and 0.88, respectively. The classification accuracy met the need of the research.

**Table 4.** Classification accuracy of LULC.

| LULC Type | 2010 | | 2020 | |
|---|---|---|---|---|
| | PA (%) | UA (%) | PA (%) | UA (%) |
| Cropland | 94.18% | 93.17% | 94.73% | 97.02% |
| Woodland | 88.00% | 78.57% | 84.00% | 72.41% |
| Grassland | 91.00% | 79.82% | 89.00% | 83.96% |
| Built-up land | 89.33% | 93.71% | 94.00% | 95.92% |
| Unused areas | 56.00% | 66.67% | 52.00% | 59.09% |
| Water area | 87.33% | 94.93% | 93.33% | 88.05% |
| OA (%) | 91.00% | | 92.50% | |
| kappa | 0.86 | | 0.88 | |

Cropland accounted for more than 70% between 2010 and 2020, but its share declined by 6.4% over the decade, a total reduction of 2593.46 ha. The proportion of built-up land increased from 15.51% to 20.74%, reaching 8375.49 ha. In the multi-scenario simulation of 2030 LULC, we found that built-up land area would increase significantly in three scenarios, which reached 10,120.32 ha in NMA. As shown in Figures 6 and 7, there would be a significant difference between NRM and NMA in cropland and water. If no reclamation measures are taken, the cropland area would be lost close to 2000 ha, while the water area

would increase by 2740 ha. Compared with NRM, the area of cropland would increase by 1013 ha, and the water area would decrease by 1217 ha.

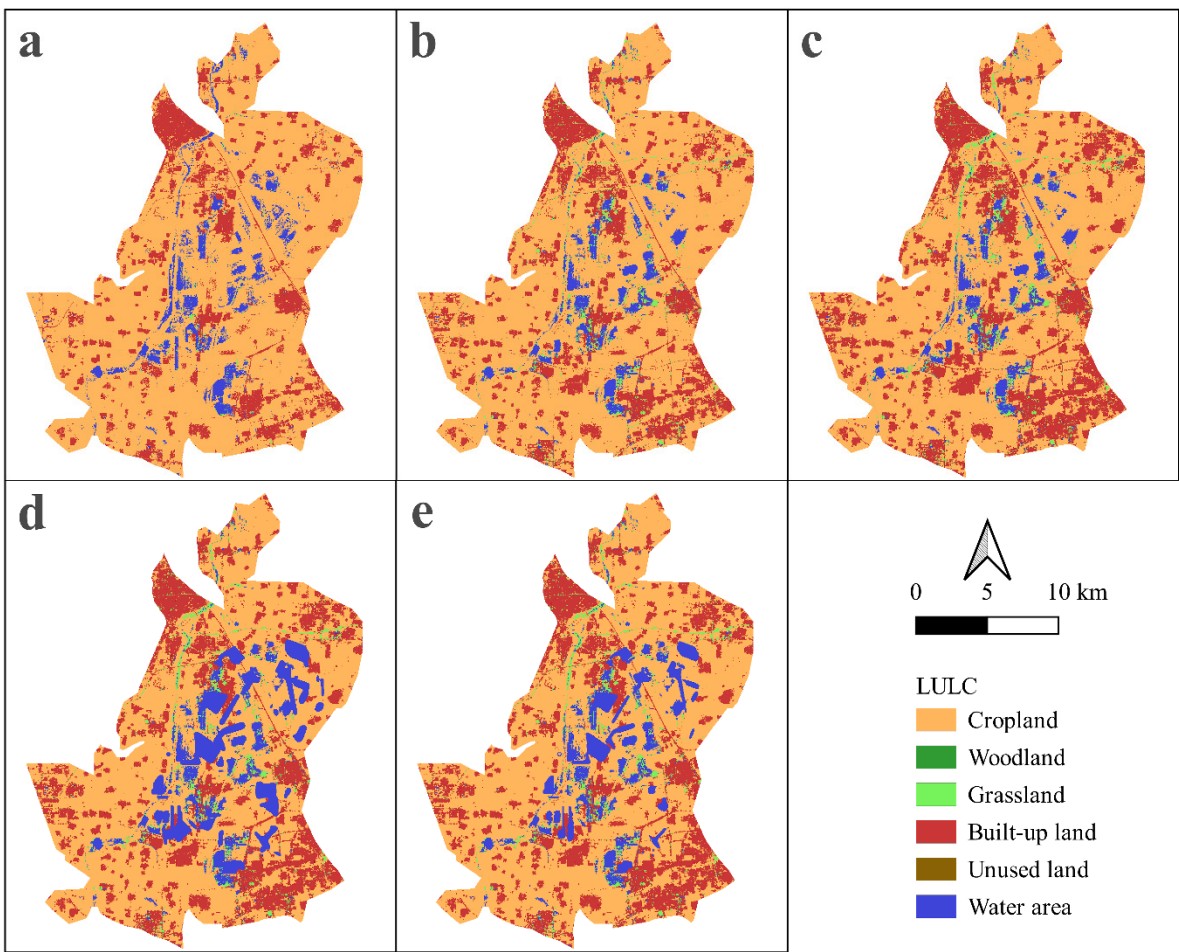

**Figure 6.** Spatial distribution of LULC: (**a**) 2010; (**b**) 2020; (**c**) NMA; (**d**) NRM; (**e**) MR.

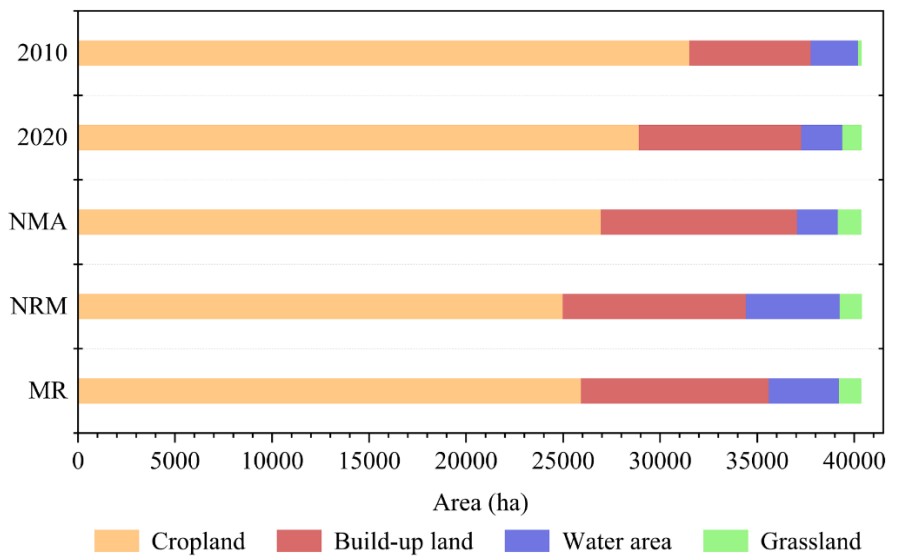

**Figure 7.** Area statistics for different LULC. (Unused land and woodland are too small to be counted.)

From the spatial distribution of different LULC, the new built-up land in 2030 would be distributed around the original built-up land. The newly added water bodies in NRM

and MR scenarios would basically be within the mining subsidence range and would mainly be areas that were originally cropland. It can be seen that mining severely impacts YZC's food security.

### 3.3. Results of Carbon Storage Estimation

From 2010 to 2020, YZC's carbon storage dropped from 1,143,225.16 to 1,099,412.47 Mg and would continue to decline in 2030 (Table 5). Even if mining were to cease in the future, the expansion of built-up land due to urban economic development would reduce 44,699.6 Mg of carbon storage in the NMA scenario. In the NRM scenario, YZC's carbon storage would only be 1,017,065.75 Mg, 82,346.71 Mg less than in 2020. In the MR scenario, the carbon storage of YZC would be 1,035,216.79 Mg, 18,151.03 Mg more than that of NRM. The average carbon density of each mine was significantly different. In 2010, the Xinglong mine with the lowest average carbon density (24.55 Mg/ha) was 6.2 Mg/ha lower than the Shanjiacun mine with the highest average carbon density (30.75 Mg/ha). Shanjiacun mine would be 9.15 Mg/ha higher than the Nantun mine in the NRM scenario. The standard deviations of average carbon density in 2010, 2020, NMA, and NRM scenarios were 1.69, 2.00, 2.35, and 2.76, respectively, indicating that coal mining leads to an increasingly apparent gap in carbon density among different coal mines.

**Table 5.** Total carbon storage and density of each mine in different periods.

| | 2010 | | 2020 | | NMA | | NRM | | MR | |
|---|---|---|---|---|---|---|---|---|---|---|
| | **Mg** | **Mg/ha** | **Mg** | **Mg/ha** | **Mg** | **Mg/ha** | **Mg** | **Mg/ha** | **Mg** | **Mg/ha** |
| Baodian | 95,097.97 | 27.05 | 92,114.36 | 26.20 | 89,675.94 | 25.50 | 79,722.99 | 22.67 | 83,969.50 | 23.88 |
| Beisu | 79,173.56 | 27.09 | 72,867.65 | 24.93 | 65,880.23 | 22.54 | 65,880.23 | 22.54 | 65,880.23 | 22.54 |
| Datong | 33,162.10 | 28.39 | 31,598.83 | 27.05 | 30,282.25 | 25.92 | 30,282.25 | 25.92 | 30,282.25 | 25.92 |
| Dongtan | 176,068.13 | 29.39 | 170,302.82 | 28.42 | 165,247.74 | 27.58 | 149,167.59 | 24.90 | 158,160.24 | 26.40 |
| Gucheng | 48,217.04 | 28.95 | 47,587.00 | 28.57 | 46,544.92 | 27.94 | 46,544.92 | 27.94 | 46,544.92 | 27.94 |
| Henghe | 29,207.00 | 27.50 | 28,543.83 | 26.88 | 27,951.31 | 26.32 | 27,451.27 | 25.85 | 27,726.96 | 26.11 |
| Liyan | 65,943.97 | 29.52 | 63,345.92 | 28.36 | 60,835.32 | 27.23 | 60,835.32 | 27.23 | 60,835.32 | 27.23 |
| Nantun | 97,611.11 | 27.40 | 88,211.17 | 24.77 | 80,763.19 | 22.67 | 77,539.11 | 21.77 | 79,869.75 | 22.42 |
| Shanjiacun | 18,436.03 | 30.75 | 18,746.12 | 31.26 | 18,538.82 | 30.92 | 18,538.82 | 30.92 | 18,538.82 | 30.92 |
| Taiping | 65,574.81 | 30.26 | 64,180.26 | 29.61 | 62,796.85 | 28.98 | 62,796.85 | 28.98 | 62,796.85 | 28.98 |
| Tianzhuang | 97,304.52 | 30.29 | 91,270.22 | 28.42 | 86,311.79 | 26.87 | 86,311.79 | 26.87 | 86,311.79 | 26.87 |
| Xingcun | 113,058.93 | 30.08 | 108,220.02 | 28.79 | 101,840.17 | 27.09 | 101,796.83 | 27.08 | 101,839.59 | 27.09 |
| Xinglong | 141,720.12 | 24.55 | 140,189.05 | 24.29 | 137,284.50 | 23.78 | 129,452.26 | 22.43 | 131,698.73 | 22.82 |
| Yangcun | 82,649.87 | 30.11 | 82,235.22 | 29.95 | 80,759.83 | 29.42 | 80,745.52 | 29.41 | 80,761.82 | 29.42 |
| Total | 1,143,225.16 | 28.31 | 1,099,412.47 | 27.23 | 1,054,712.87 | 26.12 | 1,017,065.75 | 25.19 | 1,035,216.79 | 25.64 |

We plotted the change rates of different mines in 2020 compared with 2010 and in 2030 compared with 2020. In Figure 8, we can clearly see the response of different coal mines to carbon storage change in different scenarios. From 2010 to 2020, we found that the carbon storage of the Nantun mine changed significantly compared to 2010, a reduction of 9.63%. In addition, the reduction rate of carbon storage of the Beisu mine and Tianzhuang mine was much larger than the average value of YZC. Only the Shanjiacun mine increased its carbon storage by 1.68%. In the NMA scenario, all mines' coal storage would be decreasing. The rate of change of the Datong, Tianzhuang, and Nantun mines would be lower, which may be related to the mining area's ecological background and a stronger self-recovery ability. In the NRM scenario, Baodian, Dongtan, and Nantun mine would decrease, especially in Baodian, where the change rate would be −13.45%. In the MR scenario, the carbon storage of those mining areas significantly affected by coal mining would be significantly higher than that in the NRM scenario due to reclamation work. Especially in Dongtan Mine, the reduction rate of carbon storage would be reduced from 12.41% of MR to 7.13%.

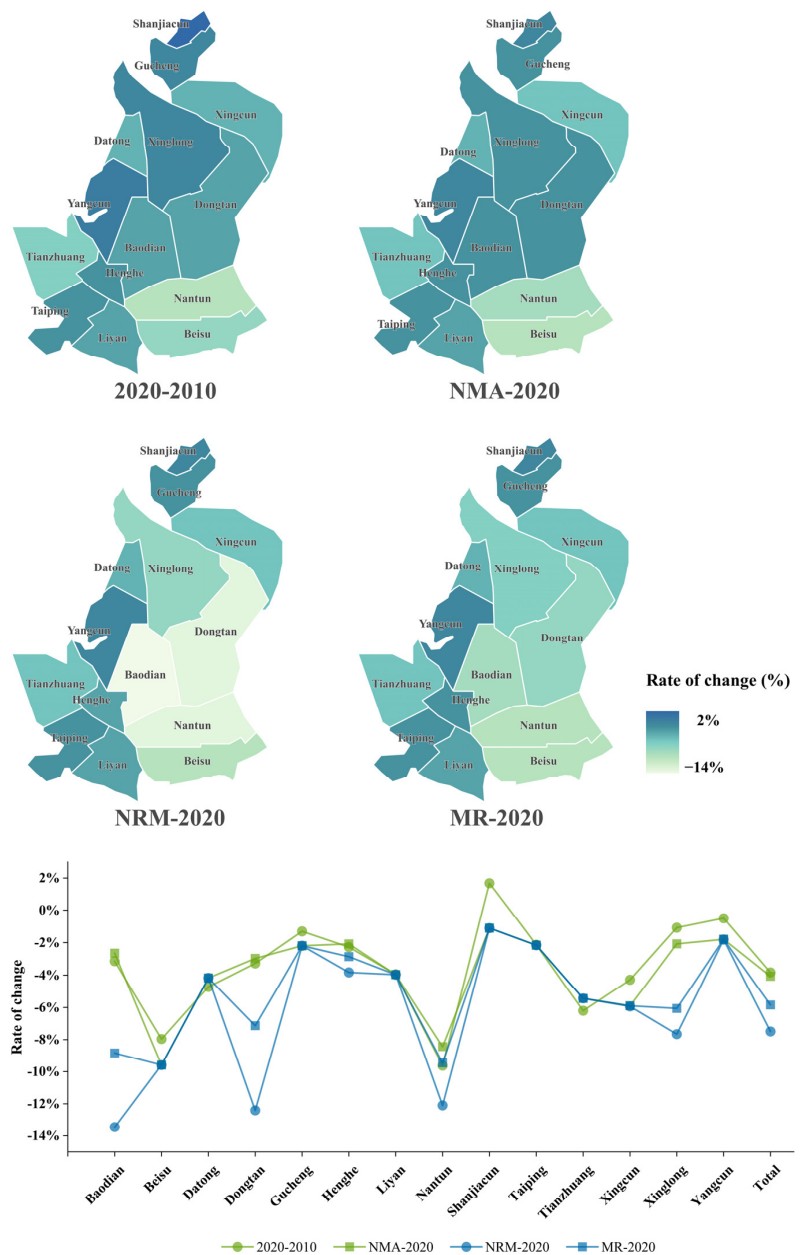

**Figure 8.** Rate of change of total carbon storage of different mines in different periods.

### 3.4. Spatial Pattern Analysis

To grasp the spatial distribution of carbon storage change more clearly and to carry out the conservation and restoration work, we carried on with the grid analysis. From 2010 to 2030, YZC has a uniform distribution of high carbon density. However, with time, the area of high carbon density would gradually decrease. In the GeoDA software, the global Moran's I index of carbon density in 2010, 2020, NMA, NRM and MR scenarios was calculated as 0.611, 0.600, 0.619, 0.618 and 0.617, respectively (Figure 9). The global Moran's I index is greater than 0 and above 0.6, indicating that the overall distribution of YZC carbon storage shows a significant spatial positive correlation.

To reveal the local spatial aggregation of carbon storage in the study area, according to the remarkable spatial aggregation relation of 5466 grid units of YZC, the local spatial autocorrelation analysis was carried out at the remarkable level of 0.001. The spatial correlation between the specific attributes of each grid and the surrounding grid elements was represented by the aggregation graph of the local indices of spatial association (LISA)

and was then divided into four types (high–high aggregation, low–low aggregation, low–high aggregation, and high–low aggregation). According to Figure 10, the distribution of YZC clusters is scattered, and the clusters are mainly high–high and low–low. As the concentrated area of the built-up area is not disturbed by coal mining, the types of clusters in the northwest are stable. However, the aggregation types in the middle east varied greatly. The enlargement of regions 1 and 2 in Figure 10f–i showed that both regions show high–high aggregation reduction and low–low aggregation increase, indicating a significant decrease in carbon storage and a decrease in carbon sink function; the ecological source of carbon storage will be gradually broken. Compared with regions 1 and 2 of Figure 10i,j, it is found that the fragmentation of carbon sequestration source areas caused by coal mining can be significantly improved by land reclamation.

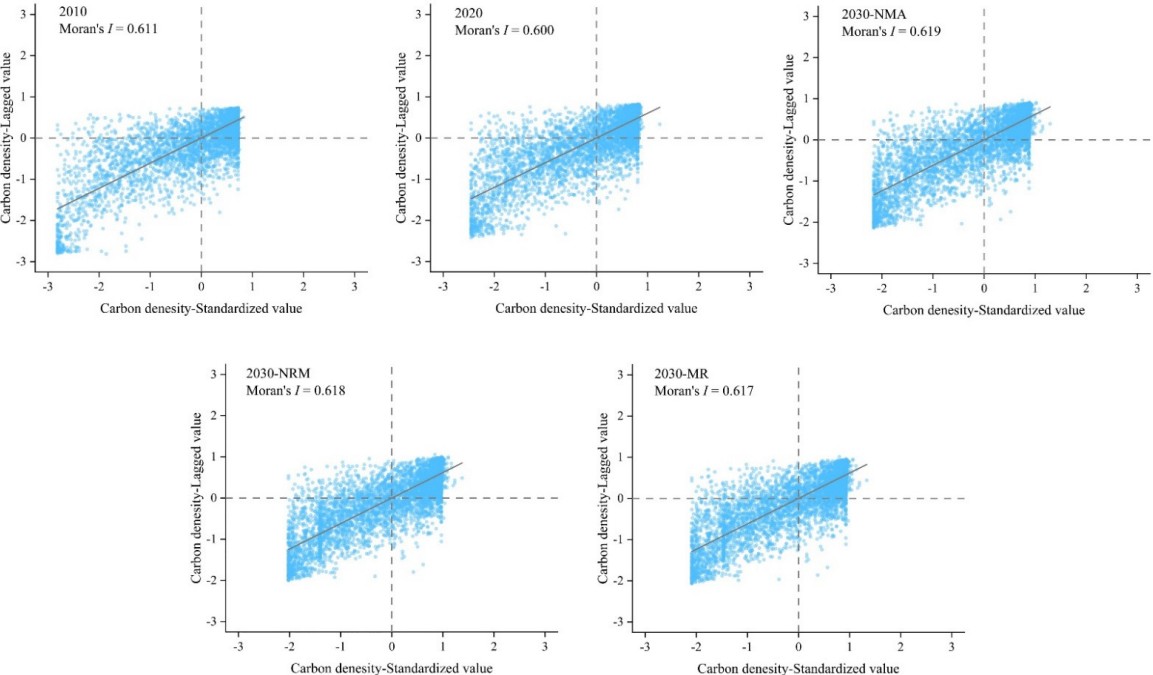

**Figure 9.** Moran's I scatterplot.

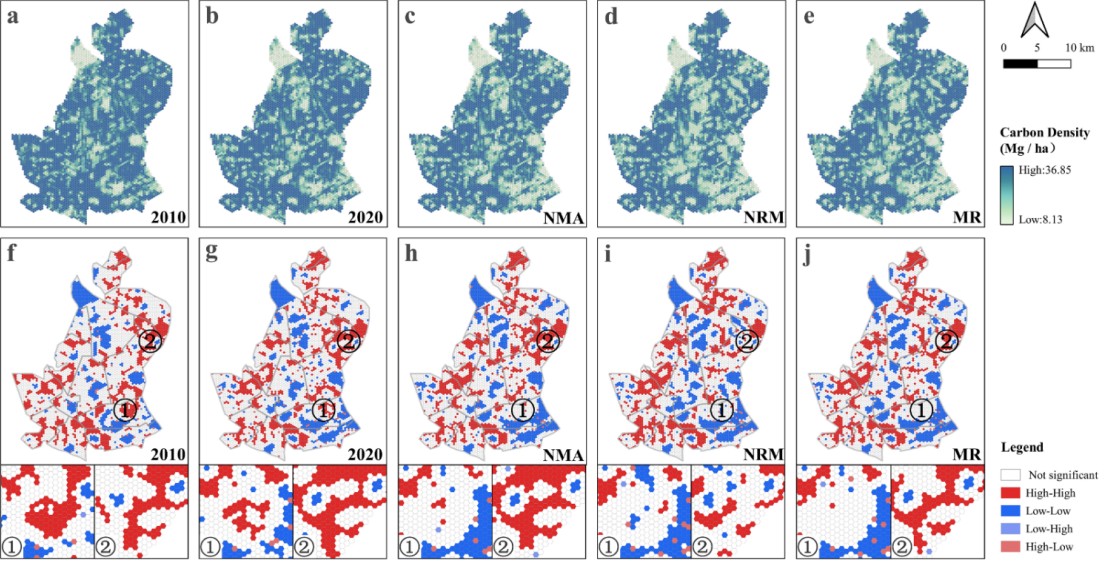

**Figure 10.** Spatial analysis of carbon storage: (**a**–**e**) carbon density; (**f**–**j**) LISA cluster map.

## 4. Discussion

### 4.1. Ecological Problems in High Submersible Coal Mining Areas in Eastern China

At present, many studies have monitored the ecological environment of high submersible mining areas in eastern China [39,40], mainly focusing on land use change [62,63], ecosystem service value assessment, water accumulation area identification [64,65], settlement monitoring [66] and vegetation quality evaluation [67,68]. This area is of concern by scholars, mainly because the mining area is located in eastern China, where the economy is more developed and densely populated. The land here carries the multiple functions of living, food production, economic construction, and coal mining, and underground coal mining seriously undermines surface ecology, production, and life.

Currently, the focus of coal resources exploitation in China has shifted from eastern China to northwestern China, but some coal mines in the eastern part will continue to produce. The existing research mainly evaluates and summarizes the ecological problems that have occurred. We used the scenario simulation method, PLUS model, and PIM to predict future surface changes to obtain LULC under various scenarios in 2030. It was found that the coal mining of YZC would continue to cause surface subsidence and form large-scale water accumulation, resulting in dramatic changes in land structure. There would be 4341.13 ha of land disturbed to varying degrees, forming more than twice the water area in 2020.

At present, China is carrying out all kinds of ecological protection and restoration to achieve the goal of carbon neutrality; thus, it is urgent to optimize the carbon accounting system. The carbon emission of underground coal mining mainly comes from fossil energy and electricity used in mining, transportation, processing, and slag emission. Carbon sink capacity mainly depends on vegetation and soil microorganisms in the mining area and on emerging CCUS technology.

Remote sensing technology will be used to monitor large-scale carbon budgets in coal mining areas. Therefore, this paper proposes a method for estimating carbon storage in coal mining areas by coupling land use simulation with the InVEST model. We predict that YZC would lose 37,647.11 Mg C due to coal mining, and the high carbon accumulation area tends to be broken. Especially in the Baodian and Dongtan mines, carbon storage of the NRM scenario would be different from that of the NMA scenario, and it is urgent to carry out targeted protection measures.

### 4.2. Land Reclamation Can Effectively Alleviate the Weakening of Carbon Sequestration Function

Over the past 30 years, China has successfully implemented many land reclamation projects in mining areas. Most of the land reclamation model is terminal treatment. That is, reclamation is carried out after land damage. Land type conversion mainly follows the principle that the reclaimed land should be preferentially used for agriculture in China's land reclamation regulations. In combination with the actual situation of YZC as a coal–grain complex area and the local reclamation plan, the reclamation objectives are set for land types with different damage levels. If the reclamation is carried out in full compliance with the target, 18,151.03 Mg C loss can be reduced, 48.21% less than in the scenario. Especially in the Nantun mine, 72.29% carbon loss can be avoided after reclamation.

Given the complexity and urgency of land reclamation in high groundwater level mining areas, scholars have put forward the technology of concurrent mining and reclamation (CMR) [69,70], which means in the process of mining, when the land is not submerged, effective reclamation measures are taken to improve the reclamation rate. Therefore, if there is CMR technology reclamation in YZC, the local carbon sequestration function will be further protected.

According to the national mine geological survey report released by China geological survey in 2016, as of 2015, 2.14 million ha had not been reclaimed in China. However, with the gradual attention paid to the ecological restoration of mining areas in recent years, more and more mining areas have been systematically treated. In March 2022, Jining City, where YZC is located, prepared the Comprehensive treatment plan for the Jining coal mining

subsidence area (2021–2030), which set the goal of completely treating all stable subsidence areas by 2030. At the same time, the focus of China's coal production has gradually shifted from the east to the west. Many coal mines in Jining have been closed in recent years. Therefore, we believe that the MR scenario will be closer to the actual situation in 2030.

*4.3. Applicability and Limitations*

As the world's largest coal producer and consumer, China vigorously carries out carbon neutralization-related work. The ecological restoration of coal mining areas cannot be ignored, which is of great significance to achieving the goal of carbon neutralization. Some scholars have now evaluated the loss of carbon sink in mining areas through field sampling [71,72]. Still, these works are time-consuming and labor-consuming, which is unsuitable for evaluating large coalfields. Using remote sensing technology can more conveniently and quickly evaluate the carbon accounting of large-scale regions. In addition to the methods used in this article, there are also many methods to evaluate the carbon sink level by remote sensing technology. The data sources include NPP [73,74], LiDAR [75], carbon satellites [76], and so on. However, these methods are mainly used to evaluate the past change trend or the current carbon storage level and cannot predict the future situation. However, the goal of carbon neutralization is the top-level goal set by the state, and coal mining is a continuous industrial activity. The management department needs to make plans in advance for facing the possible ecological problems in the mining area in the future and to realize the goal of carbon neutralization. Therefore, our method is more suitable for predicting the impact of future coal mining and reclamation on regional carbon reserves, which can provide reference data for the management department to formulate goals and plans.

This paper uses the InVEST model to evaluate carbon storage from land use and then evaluate the regional carbon sink function. Through the principle of the model, it is found that increasing the area of high carbon density land, such as multi-planting afforestation, dramatically improves the regional carbon sink level. However, since this region is a coal–grain complex area, to ensure the safety of cropland, our reclamation strategy is to increase the cropland area preferentially. Protecting arable land, improving carbon storage levels, and directly balancing economic development and coal mining is a problem that needs to be further solved.

In addition, the data for each LULC type of carbon pool used in the InVEST model is fixed. Although it has been modified using relevant climate data, the heterogeneity of the carbon density of YZC is not considered. In addition, it has been found that the light damage caused by coal mining in high groundwater level mining areas also affects carbon density. In future research, it is necessary to conduct field research and sampling and to use various remote sensing technologies to improve the accounting accuracy of the carbon budget.

## 5. Conclusions

Aiming at the disturbance of the carbon pool caused by coal mining in high groundwater level mining areas, this paper simulated and studied coupling of PIM, the InVEST model and PLUS models, and the angle of LULC change. A method for evaluating and predicting the impact of coal mining and land reclamation on carbon storage was proposed. The results showed that YZC would have 4341.13 ha of land disturbed by underground coal mining in 2030, the loss was estimated to be 37,647.11 Mg C, and the loss of 48.21% would be avoided by land reclamation. The spatial analysis of carbon storage found that YZC carbon storage has a significant positive spatial correlation. Coal mining would lead to the fragmentation of carbon sink, and reclamation can alleviate this trend.

The scheme is suitable for high groundwater level mining areas and can quantitatively calculate the carbon benefit of coal mining and reclamation. This study can provide data support for coal mining and land reclamation plans and reference the local govern-

ment's ecological supervision and regional sustainable development plan in setting carbon neutrality targets.

**Author Contributions:** Conceptualization, J.H. and Z.H.; methodology, J.H. and Z.M.; validation, J.H., Z.H. and Z.M.; formal analysis, J.H., D.Y. and J.G.; investigation, J.H., Z.H., Z.M., G.L. and S.L.; resources, J.H., D.Y. and J.G.; data curation, J.H.; writing—original draft preparation, J.H.; writing—review and editing, Z.H. and Z.M.; visualization, J.H., G.L. and S.L.; project administration, Z.H. and Z.M.; funding acquisition, Z.H. All authors have read and agreed to the published version of the manuscript.

**Funding:** This work was supported by the Major research project of Jiangsu Key Laboratory of Coal-based Greenhouse Gas Control and Utilization (No. 2020ZDZZ04B), Postgraduate Research and Practice Innovation Program of Jiangsu Province (No. KYCX19_2194), and Postgraduate Research and Practice Innovation Program of China University of Mining and Technology (No. KYCX19_2194).

**Institutional Review Board Statement:** Not applicable.

**Informed Consent Statement:** Not applicable.

**Data Availability Statement:** The data presented in this study are available on request from the corresponding author.

**Acknowledgments:** We would like to express our gratitude and respect to the editors and reviewers for their valuable comments and suggestions to improve the quality of this paper.

**Conflicts of Interest:** The authors declare no conflict of interest.

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
