# Peer review of "How to Account for Changes in Carbon Storage from Coal Mining and Reclamation in Eastern China? Taking Yanzhou Coalfield as an Example to Simulate and Estimate"

_remotesensing, doi:10.3390/rs14092014_

Round 1
Reviewer 1 Report
Thank you for the possibility of review paper entitled “How to account for changes in carbon storage from coal mining and reclamation in eastern China? Taking Yanzhou Coalfield as an example to simulate and estimate” submitted to the Remote Sensing journal. The paper deals with the important topic of carbon storage estimations of different types of terrestrial ecosystems. It focused on the different scenarios for the development of mining activities at mining region at China. The use of remote sensing approach and the InVEST model increased the quality of the paper.
The paper suits to the journal scope and is generally of high scientific quality. The spatial analysis has been conducted correctly and the method applied has been explained in details. The biggest defect deals with the discussion section and lack of the relation to law regulations on future scenarios of development.
Introduction: Lines 55-56: Describe more in details listed methods, including its advantages and disadvantages.
Methods: Why did the research deals with this particular mining region of China (outstanding features, ecological damage, water problems, etc.)
Methods: To what date refers “no mining activity” - stop of all the activity some time before 2030 or on 2030?
Methods: Unused areas can be of different LULC forms, e.g. shrubs, trees, abandoned concrete areas, barren lands. How does its type affect the carbon storage estimated in the paper?
Study area: Provide data on political and management aspects of mining activities of the study area.
Figure 2: Legend provided do not applied to all the spatial data, e.g. soil type cannot be high/low
Line 173: provide references
Discussion: Include the following data:
1) Which scenario is the most probable based on the current state, trends, demand for coal etc.
2) How does simulations corresponds to the spatial development plants and China mining policy for 2030?
3) Compare adopted method with similar methods based on the application of remote sensing.
4) How does remote sensing help to estimate carbon storage versus traditional (non-spatial) approaches?
Author Response
Thank you for your comments. We revised them according to the your opinions, please view the document.

Reviewer 2 Report
The manuscript entitled “How to account for changes in carbon storage from coal mining and reclamation in eastern China? Taking Yanzhou Coalfield as an example to simulate and estimate” focuses on a topic of sure interest for the readership of the Remote Sensing journal. However, it has some unclear issues. My main concerns are the following:
- 2.1 Why was this case study chosen? This should be well justified.
- Line 102: River River .repetition
- Line 124-125: “We used the ENVI 5.3 random forest classification tool” why did you use this classification method and not others? Please justify
- Line 125: The authors used six land use categories. Why did you use this division? All these aspects should be justified.
- Line 128: “The total accuracy of classification is 92.50%”. This should show up in the results. And what was the accuracy for each of the land use classes?
- The choice of variables presented in Table 1 should be justified and based on scientific works. Furthermore, do you not consider that the use of variables with a resolution of 1km cannot limit the interpretation of the results achieved?
- 2.3.2. PLUS model: why did you use this model and not others? This should be justified.
- Line 194 and following: once again the authors should justify why you use 2030 for the land use and land cover change projections. And why do the authors apply these specific 3 scenarios?
- 2.2.3. And why did you use the InVEST model and not other methodological approaches?
- Figure 6: Please improve the quality of these maps.
- It is necessary to validate the results achieved in section 3.2. LULC changes and simulations and 3.3 Results of carbon storage estimation.
- In the discussion, you should explain the results achieved in your study.
Author Response

(The authors gave the same response as above.)

Round 2
Reviewer 1 Report
The paper has been improved significantly according to all the reviewers comments. In my opinion, the paper can be published in present form.
Reviewer 2 Report
Thank you and your colleagues for the changes that you have made to this manuscript and how well you have answered the comments and suggestions. In my opinion, the manuscript can be published.